# Valorization of Pig Brains for Prime Quality Oil: A Comparative Evaluation of Organic-Solvent-Based and Solvent-Free Extractions

**DOI:** 10.3390/foods13172818

**Published:** 2024-09-05

**Authors:** Jaruwan Chanted, Visaka Anantawat, Chantira Wongnen, Tanong Aewsiri, Worawan Panpipat, Atikorn Panya, Natthaporn Phonsatta, Ling-Zhi Cheong, Manat Chaijan

**Affiliations:** 1Food Technology and Innovation Research Center of Excellence, School of Agricultural Technology and Food Industry, Walailak University, Nakhon Si Thammarat 80160, Thailand; jaruwanchanted@gmail.com (J.C.); pvisaka@wu.ac.th (V.A.); chantira.wo@wu.ac.th (C.W.); atanong@wu.ac.th (T.A.); pworawan@wu.ac.th (W.P.); 2Food Biotechnology Research Team, Functional Ingredients and Food Innovation Research Group, National Center for Genetic Engineering and Biotechnology (BIOTEC), National Science and Technology Development Agency, Bangkok 12120, Thailand; atikorn.pan@biotec.or.th (A.P.); natthaporn.pho@biotec.or.th (N.P.); 3School of Agriculture and Food, Faculty of Science, University of Melbourne, Parkville, VIC 3010, Australia; lingzhi.cheong@unimelb.edu.au

**Keywords:** oil extraction, pig brain, by-product, wet rendering, green extraction

## Abstract

Pig processing industries have produced large quantities of by-products, which have either been discarded or used to make low-value products. This study aimed to provide recommendations for manufacturing edible oil from pig brains, thereby increasing the value of pork by-products. The experiment compared non-solvent extraction methods, specifically wet rendering and aqueous saline, to a standard solvent extraction method, the Bligh and Dyer method, for extracting oil from pig brains. The yield, color, fatty acid profile, a number of lipid classes, and lipid stability against lipolysis and oxidation of the pig brain oil were comprehensively compared, and the results revealed that these parameters varied depending on the extraction method. The wet rendering process provided the highest extracted oil yield (~13%), followed by the Bligh and Dyer method (~7%) and the aqueous saline method (~2.5%). The Bligh and Dyer method and wet rendering techniques produced a translucent yellow oil; however, an opaque light-brown-red oil was found in the aqueous saline method. The Bligh and Dyer method yielded the oil with the highest phospholipid, cholesterol, carotenoid, tocopherol, and free fatty acid contents (*p* < 0.05). Although the Bligh and Dyer method recovered the most unsaturated fatty acids, it also recovered more trans-fatty acids. Aqueous saline and wet rendering procedures yielded oil with low FFA levels (<1 g/100 g). The PV of the oil extracted using all methods was <1 meq/kg; however, the Bligh and Dyer method had a significant TBARS content (7.85 mg MDA equivalent/kg) compared to aqueous saline (1.75 mg MDA equivalent/kg) and wet rendering (1.14 mg MDA equivalent/kg) (*p* < 0.05). FTIR spectra of the pig brain oil revealed the presence of multiple components in varying quantities, as determined by chemical analysis experiments. Given the higher yield and lipid stability and the lower cholesterol and trans-fatty acid content, wet rendering can be regarded as a simple and environmentally friendly method for safely extracting quality edible oil from pig brains, which may play an important role in obtaining financial benefits, nutrition, the zero-waste approach, and increasing the utilization of by-products in the meat industry.

## 1. Introduction

Recent increases in global meat consumption have resulted in the daily production of edible meat by-products (MBPs) from slaughterhouses despite a decline in their use for human consumption [1]. The Department of Livestock Development expects Thailand’s pig production to surpass 18.1 million heads in 2024 and pork consumption to reach 1.3 million tons [2]. The edible MBPs consist of a variety of products, including internal organs (e.g., livers, hearts, spleens, kidneys, lungs, and brains), entrails, and other parts, such as the tail, head, and feet, among others [1,3]. These edible MBPs constitute a significant ratio of the live weight of an animal, and the yields of these MBPs vary depending upon the species, ranging from 10 to 30% of the live weight of pigs and cattle [1]. The large number of MBPs has become a burden for slaughterhouses to dispose of when they are not utilized [3]. Some countries around the world traditionally use these MBPs in a variety of recipes [3]. These MBPs can effectively add value using additional processes such as centrifugation, chemical, thermal, washing, and combined processes to produce lard, gelatin, protein hydrolysates, flavor concentrate, red blood cells, plasma, and other products [4,5,6]. However, several factors, including culture, religion, income, and personal preferences, influence the consumption of MBPs. Depending on local traditions and practices, what is considered edible in one region may be viewed as inedible in another. In fact, some countries incorporate highly nutritious MBPs, such as blood, livers, hearts, lungs, spleens, kidneys, tripe, and brains, into their cuisines [3]. Although pig brains are common MBPs of slaughtering and pork processing, their widespread use, particularly for human consumption, remains unexplored.

Pig brains comprise approximately 10% fat and 8% protein, followed by carbohydrates and ash [6]. The protein isolate from pig brains was produced using the alkaline pH-shift technique, resulting in approximately a 32% dry matter yield [7]. However, the extraction of fat from the pig brain for subsequent application was not performed in a systematic manner. Pig brains provide an enormous and somewhat underutilized supply of oil that can be used to supplement human and pet diets with polyunsaturated fatty acids (PUFA). However, it is important to evaluate the lipid class of pig brains, which may contain significant levels of saturated fatty acids (SFA) and cholesterol [6]. Thus, different extraction processes may recover those lipid components to varying degrees, as eliminating such components may be required to fully examine the usage of pig brains for human nutrition. Although cholesterol and SFA fractions may be utilized in cosmetics to accomplish the zero-waste goal, these components should be avoided for intensive consumption.

The deterioration of lipids in muscle foods and animal by-products necessitated the development of an efficient and rapid total lipid extraction and purification process. Due to the highly unsaturated nature of brain lipids [6], the approach required only minimal treatment to limit oxidative deterioration and artifact formation. In addition, non-toxic and environmentally acceptable methods are required.

Rendering is the process of converting whole animal fatty tissue into concentrated fats, such as lard or tallow. There are several rendering methods available, including dry rendering, wet rendering, and frying [8]. Wet rendering can be accomplished by boiling the tissue in water or steaming it at high temperatures. Dry rendering involves cooking the fat tissues with dry heat [9]. Wet rendering is now more popular in industries than dry rendering because the resulting fat contains fewer free fatty acids (FFA) [8]. The wet rendering technique, which is green and free of hazardous compounds, might be used to extract lipids from biological sources such as fish discards, like tuna heads and eyeballs [10,11], as well as pig brains in this case, without producing environmental issues. The tuna oil industry mainly employs the wet rendering method, which comprises heating and pressing, and regards it as an environmentally friendly procedure for oil recovery [11]. Research has indicated that increasing the heating time from 10 to 30 min improved the oil yield from precooked skipjack tuna heads by 2.3% to 2.8% [10]. In order to extract the oil from the eyeballs, the blended eyeballs were mixed with distilled water (1:1, *w*/*v*) and autoclave-heated at 121 °C for 20 min, yielding 16.49% (wet weight). It is believed that heating raw material for a longer period of time helps denature proteins associated with oil, hence freeing oil to a greater extent, as demonstrated in the case of tuna eyeballs versus skipjack tuna heads [10,11]. However, high temperatures and prolonged heating can diminish the oxidative stability of the resultant oil [11]. This was in agreement with the concept that the rendering condition, particularly temperature, has a significant impact on the quality of rendered lipids. The lipids rendered at a low temperature were of higher quality than those produced at high temperatures [12].

Because of its safety and low environmental impact, aqueous oil extraction could be employed as an alternate approach for extracting oil from pig brains, as it has been performed with other raw materials [13]. Particle size reduction via procedures such as grinding, mincing, or homogenizing is typically performed in the early stages of aqueous lipid extraction from muscle foods. Subsequently, the lipids are extracted using an aqueous solution employing cellular hydrolysis with alkali, acid, or enzymes to improve the efficiency of extraction [13]. However, using a hydrolysis step limits the possible reuse of proteins and sometimes results in a bitter flavor [13]. Previous research has shown that sodium chloride can improve oil extraction from a variety of raw materials, including fish tissue, silkworm pupae, and olive paste, by altering the difference in ionic charge and density around the oil and hydrophilic phases [13,14,15]. The aqueous saline extraction method for extracting oil from pig brains has not been studied yet, but it has the potential to be an effective extraction technique.

The reference solvent extraction method employed in this study was Bligh and Dyer’s method [16]. This method extracts and purifies the total lipids in biological materials in a single step. Chloroform and methanol mixtures are widely used as lipid extractants, and an investigation of the chloroform–methanol–water phase led to the following hypothesis: When chloroform and methanol are mixed with the tissue’s water, a monophasic solution should develop, allowing for effective lipid extraction. It can then dilute the resultant homogenate with water and/or chloroform, forming a biphasic system with the lipids in the chloroform layer and the non-lipids in the methanol–water layer. Thus, separating the chloroform layer should produce a pure lipid extract [16].

The objective of this study was to compare non-solvent extraction methods, such as wet rendering and aqueous saline, to a traditional solvent extraction method, the Bligh and Dyer method, for extracting oil from pig brains. The oil yield, physical characteristics, lipid class, fatty acid composition, lipolysis, and lipid oxidation of pig brain oil were thoroughly compared.

## 2. Materials and Methods

### 2.1. Chemical

This study employed chemicals and reagents from Sigma-Aldrich (St. Louis, MO, USA), such as trichloroacetic acid (TCA), thiobarbituric acid (TBA), sodium chloride (NaCl), potassium hydroxide (KOH), 1,1,3,3-tetramethoxypropane, ammonium thiocyanate, chloroform, hexane, and methanol.

### 2.2. Pig Brains

Twenty crossbred pig brains (Landrace × Large White × Duroc, LLD), aged 4 months, were acquired from Shaw Processing Food Co., Ltd. in Nakhon Si Thammarat, Thailand. These brains were sourced from healthy pigs recognized by Thailand’s Bureau of Livestock Standards and Certification. Within an hour of collection, the samples were transported on ice to Walailak University’s Laboratory at a sample-to-ice ratio of 1:2 (*w*/*w*). Upon arrival, the brains were rinsed with cold water (4 °C), drained, and chopped using a Talsa Bowl Cutter K15e (The Food Machinery Co., Ltd., Kent, UK) to create a homogeneous composite sample. The ground samples were then vacuum-packed (DZQ-400, Afapa Vacuum Equipment Co., Ltd., Shanghai, China) and stored at −80 °C for no more than one month before experimental use.

### 2.3. Extraction of Pig Brain Oil Using Different Techniques

In this study, non-solvent extraction methods, namely wet rendering and aqueous saline, were used to extract oil from pig brains in comparison with a traditional solvent extraction method, the Bligh and Dyer method [16].

#### 2.3.1. Wet Rendering Process

Ground pig brains (500 g) were steamed in an electrical steamer (Tefal UltraCompact VC145140, Tefal, Datchet, UK) at 90–95 °C for 90 min. Due to the high moisture content of pig brain (about 80%, *w*/*w*) [6], no additional water was required in this wet rendering process. After heating, the liquid was filtered through a layer of sheet cloth, and the pig brain leftovers were removed. Following allowing the liquid to cool to room temperature (27–29 °C), the fat layer was collected using a separatory funnel into a 125 mL Erlenmeyer flask containing 2–4 g of anhydrous sodium sulfate for removing the remaining water. The flask was then thoroughly shaken and decanted through Whatman No. 4 filter paper. The procedure took roughly 100 min in total. The moisture content of the oil sample was determined using a coulometric Karl Fischer titrator (C20, Mettler-Toledo Intl., Columbus, OH, USA). The oil was stored in an amber vial at −20 °C in a nitrogen environment until analysis.

#### 2.3.2. Aqueous Saline Process

The aqueous saline extraction method was modified from the work of Tzompa-Sosa et al. [17] and Kadioglu et al. [18]. To extract the oil, 100 g of ground pig brain was combined with 300 mL of saline (1.7% *w*/*v*, NaCl). After blending for 3 min using a Phillips HR2118 food processor (Phillips International Co., Ltd., Samut Prakan, Thailand), the mixture was stirred (100 rpm/30 min) with an IKA^®^ Rw 20 digital overhead stirrer (Staufen, Germany). To remove big particles, the mixture was then filtered using a 45-mesh stainless-steel screen.

The pig brain suspension was centrifuged (RC-5B plus centrifuge, Sorvall, Norwalk, CT, USA) at 15,000× *g* for 1 h at 25 °C to separate it into 3 fractions: cream, supernatant or aqueous layer, and pellet (top to bottom). For better lipid separation, the top cream phase and supernatant layer were collected and centrifuged at 20,000× *g* for 20 min at 25 °C. Following the second centrifugation, free oil was located in the top layer of the centrifuge tube. The extracted oil was then transferred to a 125 mL Erlenmeyer flask containing 2–4 g of anhydrous sodium sulfate to remove any remaining water, using the same approach as described above for the wet rendering process. In total, this procedure took about 120 min. After the moisture content was determined, the oil was also stored in an amber vial at −20 °C in a nitrogen atmosphere until analysis.

#### 2.3.3. Bligh and Dyer Process

The Bligh and Dyer method [16] was used as the conventional method for extracting total lipids from ground pig brains. The samples (25 g) were homogenized with 200 mL of a combination of chloroform, methanol, and distilled water (1:2:1, *v*/*v*/*v*) at 9500 rpm for 2 min at 4 °C using an Ultra Turrax homogenizer (Ultra Turrax IKA T18 basic, Wilminaton, NC, USA). The homogenate was mixed with 50 mL of chloroform and homogenized for 1 min. Then, 25 mL of distilled water was added, and the mixtures were homogenized at the same speed for 30 s. The homogenate was centrifuged at 3000× *g* at 4 °C for 15 min (RC-5B plus centrifuge, Sorvall, Norwalk, CT, USA) and placed into a separating flask. The chloroform phase was drained into a 125 mL Erlenmeyer flask containing ~5 g of anhydrous sodium sulfate, agitated thoroughly, and decanted into a round-bottom flask using Whatman No. 4 filter paper. Following that, the solvent was removed at 40 °C with an Eyela rotary evaporator (Model N-100, Tokyo Rikakikai, Co. Ltd., Tokyo, Japan). The entire process took about 45 min. Then, the moisture content was determined. The oil was likewise kept in an amber vial at −20 °C in a nitrogen environment until analysis.

### 2.4. Determination of Extraction Yield

The yield was calculated using Equation (1), which employed the total weight of extracted oil after subtracting moisture content obtained using a coulometric Karl Fischer titrator.
Extraction yield (%) = (Total weight of extracted oil − Moisture/Sample weight) × 100(1)

### 2.5. Color Analysis

Colorimetric values of pig brain oils, including *L** (lightness), *a** (redness/greenness), and *b** (yellowness/blueness) were analyzed using a Hunterlab ColorFlex^®^EZ instrument (10° standard observers, illuminant D65; Hunter Assoc. Laboratory; Reston, VA, USA). Before performing color analysis, the equipment was calibrated using white and black standard plates. A glass sample cup was filled with oil samples of equal weight from each treatment and then analyzed for color. The redness index (*a**/*b**) was also reported.

### 2.6. Determination of Total Phospholipid (PL), Cholesterol, Carotenoid, and Tocopherol Contents

The PL content was determined using Stewart’s method [19]. The 0.01 g lipid sample was combined with 2 mL of chloroform and 1 mL of thiocyanate solution (0.1 M ferric chloride hexahydrate and 0.4 M ammonium thiocyanate). The mixture was properly mixed before being centrifuged at 1000× *g* for 5 min. The top phase’s absorbance was measured at 488 nm using a UV-vis spectrophotometer (Shimadzu, Kyoto, Japan). A standard curve was created with phosphatidyl choline values ranging from 0 to 500 ppm. The PL content was calculated as g/100 g lipid.

The total cholesterol was determined using Beyer and Jensen’s method [20]. The oil sample (0.1–0.2 g) was saponified with 2% alcoholic KOH for 10 min. The unsaponified fraction was extracted using 2 × 10 mL hexane. The extracts were rinsed with 5 mL of distilled water and dried in a water bath at 45 °C. Then, the dried extract was resuspended in 3 mL glacial acetic acid, followed by 2 mL of coloring reagent. The coloring reagent was generated by diluting 1 mL of the stock reagent with 100 mL of concentrated H_2_SO_4_ after making a stock reagent of 10% (*w*/*v*) FeCl_3_.6H_2_O in glacial acetic acid. A UV-vis spectrophotometer was used to measure the absorbance of the reaction mixture at 565 nm in relation to a glacial acetic acid blank. Cholesterol (Sigma Aldrich) in glacial acetic acid at 0–120 mg/L was used to create a standard curve. The total cholesterol was reported as mg/100 g lipid.

The total carotenoid content was assessed using the method developed by de Carvalho et al. [21]. The lipid sample (1 g) was combined with a 1:10 (*w*/*v*) organic solvent mixture (15:75:10; *v*/*v*/*v*), then homogenized at room temperature for 2 min. After 10 min at room temperature, the mixture was centrifuged (3000× *g*/25 °C/30 min) using an RC-5B plus centrifuge (Sorvall, Norwalk, CT, USA). Following that, the supernatant was filtered using Whatman No. 1 filter paper, and the volume was made up of petroleum ether. The samples were then read at 470 nm using a Shimadzu UV-vis spectrophotometer (Kyoto, Japan). The total carotenoid content was computed using the extinction coefficient of β-carotene in petroleum ether (A_1cm_) of 2400 and reported as mg β-carotene equivalents/100 g lipid.

The total tocopherol was measured using the technique of Kayden et al. [22]. The 1 g lipid sample was combined with 1 mL of absolute ethanol, followed by 0.2 mL of 0.2% bathophenanthroline in absolute ethanol. The mixture was added with 0.2 mL of absolute ethanol containing 0.001 M FeC1_3_ and then incubated at room temperature for 1 min. Following that, 0.2 mL of 0.001 M H_3_PO_4_ in absolute ethanol was added. Thereafter, the absorbance of each solution was measured at 534 nm. To produce the standard curve, α-tocopherol was used at concentrations ranging from 0 to 50 mM. Total tocopherol was measured as mg α-tocopherol equivalents/100 g lipid.

### 2.7. Determination of Fatty Acid Profiles

Fatty acid methyl esters (FAME) in the samples were measured with a gas chromatography/quadrupole time of flight (GC/Q-TOF) mass spectrometer (GC 7890B/MSD 7250, Agilent Technologies, Santa Clara, CA, USA) connected to the PAL auto sampler system (CTC Analytics AG, Zwingen, Switzerland). MS data were collected using Agilent Technologies’ MassHunter software (version 10.0, Santa Clara, CA, USA). Myristic acid D27 (500 ppm in hexane) was used as an internal standard. The calibration curves were created by combining an equivalent amount of FAME (20 to 1000 ppm) with a solution of myristic acid-D27 methyl ester in hexane. The complete procedure, as well as the optimal analytical condition, can be retrieved from Chinarak et al. [23].

### 2.8. Determination of Lipolysis and Lipid Oxidation

Lipolysis, or lipid hydrolysis, was monitored by measuring the FFA content using the method described by Lowry and Tinsley [24]. A lipid sample (0.1 g) was combined with 5 mL of isooctane, followed by the addition of 1 mL of cupric acetate-pyridine solution (5% *w*/*v*). The mixture was then vortexed for 90 s and incubated at 40 °C for 20 s. The upper phase was measured using a UV-vis spectrophotometer at 715 nm (Shimadzu, Kyoto, Japan). A standard curve was generated using oleic acid, and the FFA content was calculated and reported as g/100 g of lipid.

The formation of primary lipid oxidation products was monitored using peroxide value (PV), according to the method described by Low and Ng [25]. The sample (1 g) was treated with 25 mL of chloroform and acetic acid mixture in a 2:3 ratio. The mixture was shaken vigorously, and then 1 mL of saturated potassium iodide was added. After keeping the mixture in the dark for 5 min, 75 mL of distilled water was added. As an indicator, 0.5 mL of 1% (*w*/*v*) starch solution was added. The PV was measured by titrating the iodine liberated from potassium iodide with a standardized 0.01 N sodium thiosulfate solution. The PV was reported as milliequivalents active oxygen (meq)/kg.

The formation of secondary lipid oxidation products was determined using the thiobarbituric acid reactive substances (TBARS) assay [26]. A sample (0.5 g) was combined with 2.5 mL of a TBARS solution consisting of 0.375% TBA, 15% TCA, and 0.25 N HCl. The mixture was then heated in a boiling water bath for 10 min to produce a pink color. After cooling under running tap water, the mixture was centrifuged at 5000× *g* for 10 min at 25 °C. The absorbance of the resulting supernatant was measured at 532 nm. A standard curve was established using 1,1,3,3-tetramethoxypropane at concentrations ranging from 0 to 10 ppm, and TBARS values were reported as mg of malondialdehyde (MDA) equivalents/kg.

### 2.9. Fourier Transform Infrared (FTIR) Spectroscopy

The FTIR analysis of pig brain oil was performed following the method reported by Chaijan and Panpipat [27]. A horizontal attenuated total reflectance (ATR) trough plate crystal cell (45° ZnSe; 80 mm in length, 10 mm in width, and 4 mm in thickness) from Pike Technology, Inc. (Madison, WI, USA) was utilized in conjunction with a Bruker Model Vector 33 FTIR spectrometer (Bruker Co., Ettlingen, Germany). FTIR spectra were captured at room temperature using 16 scans at 4 cm^−1^ resolution in the mid-infrared range (4000–500 cm^−1^). A baseline was established by collecting reference air spectra. To achieve proper spectrum interpretation, this study employed automatic baseline correction techniques during the FTIR analysis. The spectral data were analyzed using the OPUS 3.0 collection software.

### 2.10. Statistical Analysis

This study employed a completely randomized design (CRD). All experiments were run in triplicate (*n* = 3). The data were analyzed using ANOVA, and the means were compared using Duncan’s multiple range test. Statistical analysis was performed with SPSS 23.0 (SPSS Inc., Chicago, IL, USA).

## 3. Results and Discussion

### 3.1. Extraction Yield

Table 1 shows the oil extraction yield from pig brains utilizing various extraction methods. The wet rendering process provided the highest extracted oil yield in the pig brain sample, accounting for 13.09%, followed by the Bligh and Dyer method (6.61%) and the aqueous saline method (2.43%). The higher the temperature in the wet rendering procedure, the more oil may be extracted from the brain matrix. It has been found that during heating, proteins denature, releasing oil from blended raw materials [11]. Tuna eyeball extraction yielded 16.49% after 20 min of autoclaving at 121 °C [11]. The greater extraction temperature of 121 °C autoclaving in tuna eyeball extraction compared to 90–95 °C steaming in this study may explain the higher yield. A low-temperature rendering approach has also been reported for recovering antioxidants from fish waste (*Decapterus maruadsi*). The process was carried out for 90 min at 30 °C with a sample-to-solvent loading ratio of 3:10 (*w*/*v*) [28].

A low extraction yield of silkworm pupae oil was also recorded using the aqueous saline technique. Silkworm pupae oil was extracted using 1.7% *w*/*v* saline solution, 1:3.3 silkworm pupae-to-aqueous liquid ratio (*w*/*v*), and 119 min stirring time at 100 rpm at 25 °C, yielding simply 3.32% oil [13]. In the case of pig brain, the structure and composition may be more complex than silkworm pupae, and these characteristics may influence the effectiveness of oil extraction from pig brain. The brain is a complex temporal and spatial multiscale structure that produces complicated molecular, cellular, and neuronal phenomena [29].

Basically, the non-polar solvent proved more effective at dissolving and extracting oil [11]. However, heating can weaken the interaction between oil and the protein matrix in the brain. This can help extract oil from the brain. Nonetheless, heat treatment has the potential for non-toxic oil extraction and is regarded as a green process [11,30].

### 3.2. Color

The appearance and color (*L**, *a**, and *b** values) of pig brain oil obtained using various extraction methods are depicted in Figure 1 and Table 1. The Bligh and Dyer method and wet rendering procedures produced a translucent yellow color with varying shades; however, an opaque brown-light red color was detected in oil extracted using the aqueous saline process (Figure 1). The highest *L** value was discovered in oil extracted using the Bligh and Dyer process, followed by aqueous saline and wet rendering (*p* < 0.05). All extraction methods produced negative *a** values in all samples, but only the aqueous saline extraction method detected a negative *b** value, resulting in a positive redness index (*a**/*b**), as seen by the redder color in Figure 1. Aqueous saline may co-extract certain red heme pigments, which are abundant in pig brain, into the oil. Chanted et al. [6] discovered that the pig brain had 1.31 g/100 g of total heme protein. This heme protein may be leached into the salt solution-pig brain homogenate, which may be contaminated by residual water present in the finished oil. Furthermore, the lipid-soluble brown pigment formed by heme protein oxidation, as well as the Maillard reaction during extraction [10], may be responsible for the discoloration of the oil extracted using the aqueous saline method. It has been noted that the *L** value of oil decreases with increasing heating time [11] since wet rendering was conducted with steaming, which might result in a reduction in the *L** value with the highest negative *a** value.

### 3.3. Total PL, Cholesterol, Carotenoid, and Tocopherol Contents

The brain is high in PL, which are crucial molecules that create the membrane lipid bilayers of neurons, glia, and cerebrovascular cells, as well as providing structural integrity for intracellular and cell surface membrane proteins [31]. In another approach, PL is known as the fundamental component of biological membranes and as a natural surfactant with high biocompatibility [32]. According to Lu et al. [33], PL has high emulsifying capabilities and might be used as a natural surfactant for emulsion formation. Also, PL has antioxidant properties [34]. The total PL concentrations of pig brain oils extracted by various techniques ranged from 0.1 to 3.2 g/100 g lipid (Table 2). The findings were consistent with those published by Chinarak et al. [35], who discovered 2.6–9.3 g/100 g of PL concentration in alternative oil samples recovered from sago palm weevil (*Rhynchophorus ferrugineus*) larvae. Pig brain oils had a lower PL concentration compared to krill oil (32.5 g/100 g lipid) [36] and mantis shrimp oil (40.6–54.0 g/100 g lipid) [37]. The Bligh and Dyer method extracted the oil with the highest PL concentration (*p* < 0.05). The wet rendering and aqueous saline techniques extracted the oil with the same PL content (*p* > 0.05), which was rather low (about 0.1 g/100 g lipid). This was most likely caused by the different solvents employed to extract the oil. Because the Bligh and Dyer method is a total lipid recovery method, the solvent utilized can recover the majority of the lipid classes. However, the toxicity of the solvents utilized can be a difficulty for edible oil production. Thus, the alternate extraction procedure without solvent, such as wet rendering and aqueous saline, used herein can recover some PL.

Cholesterol is another structural component of cell membranes that helps to synthesize vitamin D, steroid hormones, and bile acids. Aside from its structural role in maintaining stability and fluidity, cholesterol is also important in regulating cell activity [38,39,40]. From a nutritional standpoint, cholesterol is essential for good health at a reasonable amount. However, a high cholesterol intake has been connected to a variety of health problems, including hypercholesterolemia, cardiovascular problems, and coronary heart disease [41]. Adult population recommendations propose a maximum cholesterol intake of 300 mg per day [42]. The crude lipids isolated from the pig brain included cholesterol at levels of 45.6–4305.7 mg/100 g lipid. The Bligh and Dyer process yielded the oil with the highest cholesterol concentration, outperforming the aqueous saline and wet rendering extraction procedures (*p* < 0.05) (Table 2). It has been found that the cholesterol levels in cow brains (1014 mg/100 g lipid) [43], calf brains (1810 mg/100 g lipid) [44], and rabbit brains (2295.5 mg/100 g lipid) [45] are all high. When compared to other pork products, pig brain oil contained more cholesterol than lard (143 mg/100 g lipid) [46] and pork fat (131 mg/100 g lipid) [47]. Although pig brain lipids are rich in PL, it is important to consider the cholesterol concentration when ingesting pig brain. Thus, separating cholesterol from pig brain oil could help increase the oil’s utility as a functional ingredient. Unrefined pig brain oil, on the other hand, may be employed as a cosmetic ingredient because cholesterol is known to be used as an emulsifier in cosmetic skin and hair care products, as well as eye and face makeup formulas at appropriate levels [48]. Cholesterol can also be employed to induce bilayer formation in niosomes for cosmetic purposes [49].

Carotenoids are the most common tetraterpene pigments found in nature, including photosynthetic bacteria, certain archaea and fungi, algae, plants, and animals [50]. The structure of carotenoids influences their physical qualities, chemical reactivity, and biological roles. Carotenoids, in particular, aid in antioxidant capacity by sequestering singlet oxygen and scavenging free radicals [51]. Carotenoids can be discovered in pig brain oils, with concentrations ranging from 0.02 to 0.09 mg/100 g lipid in the samples, depending on the extraction method (Table 2). The Bligh and Dyer extraction method produced oil with considerably higher carotenoid content than the aqueous saline and wet rendering extraction methods (*p* < 0.05). The presence of a particular carotenoid led the pig brain oil to become yellow, as depicted in Figure 1. However, pig brain oils contain fewer carotenoids than oil from other sources such as Pacific white shrimp (*Litopenaeus vannamei*) hepatopancreas (160–180 mg/100 g lipid) [52] and sago palm weevil larvae (0.7–0.9 mg/100 g lipid) [35].

Tocopherols are the principal lipid-soluble antioxidants in the cell’s antioxidant system, protecting PUFA, low-density lipoproteins, and cell membrane components against free radical oxidation. Tocopherols are largely found in the PL bilayer of cell membranes [53]. The total tocopherol content of pig brain oil was 66.2 mg/100 g lipid from the Bligh and Dyer method, 33.1 mg/100 g lipid from the wet rendering method, and 27.4 mg/100 g lipid from the aqueous saline method (Table 2). The oil’s total tocopherol concentration varied depending on the extraction method. The Bligh and Dyer method employs solvents, which allow the oil-soluble tocopherol to be released more effectively in the presence of the solvent with optimum polarity. The tocopherol content of pig brain oil was comparable to that of mantis shrimp oil (40.6–49.7 mg/100 g lipid) [37], but it was higher than that of sago palm weevil larvae oil (18.8–22.2 mg/100 g lipid) [35]. Tocopherols have antioxidant activity because of their propensity to donate phenolic hydrogen atoms to peroxyl radicals. The tocopheroxyl radical formed is stable and will not persist in the peroxidation cycle. Alternately, it interacts with another peroxyl radical, yielding a non-radical product [54].

### 3.4. Fatty Acid Profiles

The fatty acid profiles of pig brain oil extracted using various methods are given in Table 3. The fatty acid contents vary depending on the extraction process. The Bligh and Dyer method recovered the most fatty acids because it used solvent extraction to separate the majority of the lipids from brain tissue. The wet rendering process yielded the highest total SFA concentration, followed by aqueous saline and the Bligh and Dyer method. Stearic acid and palmitic acid were the most abundant saturated fatty acids in pig brain oil recovered using all methods.

The Bligh and Dyer method retrieved the largest number and content of unsaturated fatty acids (MUFA and PUFA), followed by the aqueous saline and wet rendering methods. The Bligh and Dyer extraction has long been recognized as an effective method for extracting triacylglycerols, FFA, and PL [16], making it possible to recover more unsaturated fatty acids from pig brain. Simultaneously, the Bligh and Dyer method recovered more PUFA while concurrently extracting more trans-MUFA. The Bligh and Dyer method and the aqueous saline method recovered around 20% elaidic acid, a trans-fatty acid. However, the concentration of elaidic acid was reduced by approximately 50% in oil produced through the wet rendering method. Elaidic acid, an unsaturated trans-fatty acid, has received attention because it is a significant trans-fat identified in hydrogenated vegetable oils, as well as in trace levels in caprine and bovine milk and various meats [55]. Trans fats have been linked to heart disease [56].

Only linoleic acid (*cis*-9,12-octadecadienoic acid; C18:2 *n*-6) was discovered as a PUFA in oil extracted using wet rendering and aqueous saline techniques, and its quantity was higher than that of oil extracted using the Bligh and Dyer method. The oil extracted using the Bligh and Dyer method produced a significant amount of docosahexaenoic acid (DHA) (13.21%). The results aligned with Bruce et al. [57], who revealed that DHA is the primary PUFA in the brain. Long-chain *n*-3 fatty acids, including eicosapentaenoic acid (EPA) and DHA, can prevent cardiovascular disease, reduce inflammation, and promote brain development [58].

### 3.5. Lipolysis and Lipid Oxidation 

The hydrolysis of triglycerides and PL in oil produces FFA. It is considered an important indicator of oil rancidity and stability [59]. The high concentration of FFA in animal oils lowers their commercial value. Figure 2a depicts the FFA content of pig brain oil recovered by different methods. The highest FFA content was found in pig brain oil extracted using the Bligh and Dyer method (5.76 g/100 g oil) (*p* < 0.05), while aqueous saline and wet rendering methods produced very low FFA (<0.65 g/100 g lipid). The maximum level of FFA is up to 0.65% or 1.3 mg KOH/g fat for lard or rendered pork fat [60]. Thus, aqueous saline and wet rendering procedures are suitable for the extraction of high-quality pig brain oil with low FFA levels.

Pig brain lipids may be susceptible to oxidation because of their high PUFA content (Table 3). PV is linked to the generation of peroxides in unsaturated fat during oxidation, which is triggered by the breaking of double bonds and results in short-chain volatile molecules that cause the rancid odor. The number of peroxides in edible oil shows its oxidative level and, as a result, its susceptibility to oxidative rancidity. Oils with high PV (>10 milliequivalents active oxygen (meq)/kg) are unstable and readily become rancid, whereas oils with low PV (<10 meq/kg) are stable against oxidation [61]. The maximum PV level is 10 meq/kg fat for lard or rendered pork fat [60]. The PV of the pig brain oil extracted using all methods was less than 1 meq/kg (Figure 2b), indicating high quality in terms of the formation of primary lipid oxidation products. The Bligh and Dyer extraction process yielded the oil with the lowest PV (0.10 meq/kg), followed by wet rendering (0.52 meq/kg) and aqueous saline (0.77 meq/kg) (*p* < 0.05). In the aqueous saline extraction, long-term stirring in water (30 min) followed by twice centrifuging for a total of 80 min may enhance peroxide production as compared to the Bligh and Dyer method, which used low temperature and short period homogenization (total of 3.5 min). During wet rendering, heating for 90 min may facilitate the formation of peroxide.

TBARS is an extensively used biomarker of lipid oxidation, particularly in meat and fish products [62]. Figure 2c shows the TBARS values of pig brain oil extracted using various methods. Pig brain oil extracted using the Bligh and Dyer method exhibited a greater TBARS content (7.85 mg MDA equivalent/kg) compared to aqueous saline (1.75 mg MDA equivalent/kg) and wet rendering (1.14 mg MDA equivalent/kg) (*p* < 0.05). A high TBARS value with a low PV of oil extracted by the Bligh and Dyer method could be attributed to increased oxidation, in which the primary lipid oxidation products (i.e., hydroperoxides) were changed to secondary lipid oxidation products (i.e., aldehydes), resulting in higher TBARS. A TBARS content of 1 mg of MDA equivalent/kg shows lipid oxidation, which causes a rancid odor and taste detectable to consumers. TBARS levels of more than 2 mg of MDA equivalent/kg imply that the product is likely to be detected as rancid off-flavor by consumers while also causing other aberrant odors [63]. According to Domínguez et al. [64], meat and meat products should have no more than 2–2.5 mg MDA equivalent/kg to avoid rancidity. Overall, based on the lipolysis and lipid oxidation indices, the non-solvent approach, particularly wet rendering, can be considered to be the most successful method for pig brain oil to minimize lipid instability.

### 3.6. FTIR Spectra

Figure 3 presents the FTIR spectra of pig brain oil extracted using different methods. The oil exhibits characteristic fingerprint spectra in the wavenumber range of 3500 to 2700 cm^−1^, corresponding to the stretching vibrations of CH, NH, and OH groups, which are associated with its chemical components [65]. The peaks between 3000 and 2850 cm^−1^ are attributed to the C–H stretching bonds of methyl and methylene groups. The bands within the 3100–2750 cm^−1^ range, dominated by C–H stretching vibrations, are indicative of the fatty acid content in the lipids [66].

The peaks within the 1800-to-600 cm^−1^ range of pig brain oils extracted by different methods varied, reflecting differences in lipid purity and composition, including triglycerides, PL, cholesterol, sphingomyelin, and cerebrosides. The 1800–700 cm^−1^ interval is effective for distinguishing lipids since it serves as a unique fingerprint for each component. Bands are commonly attributed to the four-membered aromatic cholesterol ring, the glycerol backbone of PL, the ceramide backbone of sphingolipids, the hydrophilic head group, and the deformation vibrations of C–H groups in hydrophobic fatty acid chains [66].

The C–H stretch vibrations in the spectrum of cholesterol are centered around 2931 cm^−1^, with the peak identified as about 2926–2925 cm^−1^ in this work. The peak was the sharpest in the oil extracted by the Bligh and Dyer method, which contained the highest cholesterol level (Table 1). Lower wavenumber bands correspond to C–H deformation vibrations and cholesterol-specific vibrations (1464, 1378, and 754 cm^−1^). These peaks are consistent with Dreissig et al.‘s findings for pure cholesterol [66]. According to another study, the cholesterol spectrum exhibits many distinct bands ranging from 500 to 1200 cm^−1^, with the strongest ones at 754 cm^−1^ [65], as shown in this study for oil extracted using the Bligh and Dyer method. Further bands in cholesterol ester have been reportedly assigned to the C=O stretch vibration (1740 cm^−1^) and the fatty acid chain (2917, 2850, and 1179 cm^−1^), where a tiny peak at 1733 cm^−1^ was observed in oil extracted using the Bligh and Dyer method.

Based on the findings of Dreissig et al. [66], bands at 1645 and 1545 cm^−1^ are caused by an amide group made up of C=O and N–H that is present in the ceramide backbone of the sphingolipids galactocerobroside, sphingomyelin, and sulfatide. Proteins also contain bands at comparable locations because of peptide bonding. Within galactocerobroside, the C–O–H vibrations of the sugar moiety are attributed to bands at 1086, 1042, and 1017 cm^−1^. Sphingomyelin is a member of the sphingophospholipid class. It is made up of a phosphatidylcholine residue connected to a ceramide backbone. The characteristic of sulfatide is the presence of a sulfate group linked to the sugar ring, denoted by bands at 1243 and 1070 cm^−1^. When compared to the pure chemicals utilized in the literature [66], several bands linked to those compounds were discovered here; however, they might not be precisely the same wavenumber because of the oil’s purity.

Phosphatidic acid is the parent molecule for PL that lacks an alcohol head group. The spectrum of phosphatidic acid shows bands owing to C=O (1740 cm^−1^), C–H groups (1466, 1381, 1174, and 720 cm^−1^), and the phosphate group (1104 and 1074 cm^−1^). The alcohol head groups linked to the phosphate moiety alter the FTIR spectrum. The antisymmetric stretch vibration peaks of PO^2−^ are observed at 1231 cm^−1^ for phosphatidylinositol (PI), 1225 cm^−1^ for phosphatidylethanolamin (PE), and 1240 cm^−1^ for phosphatidylcholine (PC) and phosphatidylserine (PS). The locations of the symmetric stretch vibration peaks of PO^2−^ vary according to the type of alcohol group. They overlap with alcohol group bands to complicated envelopes, with prominent bands at 1104, 1074, 1039, and 862 cm^−1^ (PI), 1073 and 1030 cm^−1^ (PE), 1090, 970, and 827 cm^−1^ (PC), and 1097 and 1058 cm^−1^ (PS). Again, multiple PL-related bands were observed herein, but they may not be exactly the same wavenumber due to the oil’s purity when compared to the pure compounds employed in the literature [66].

The vibratory OH stretching of water causes the first band (3600–3200 cm^−1^) to function, with the bands being larger in oil from wet rendering and aqueous processes due to the existence of higher residual moisture content than oil recovered by the Bligh and Dyer process. According to Dreissig et al. [66], the peak at 1637 cm^−1^ corresponds to the O–H deformation vibrations of water impurities. This peak was higher in oil samples from both wet rendering and aqueous saline treatments.

Overall, FTIR spectra of pig brain oil indicated the existence of numerous components with varied amounts as determined by chemical analysis experiments.

## 4. Conclusions

The method of oil extraction from pig brains significantly influences the yield, composition, and stability of the oil. Wet rendering provided the highest oil yield, while the Bligh and Dyer method produced oil with the highest content of phospholipids, cholesterol, carotenoids, tocopherols, and unsaturated fatty acids, albeit with a higher trans-fatty acid content. The aqueous saline method resulted in the lowest yield and produced a discolored oil. Despite yielding the highest quality oil in some respects, the Bligh and Dyer method also led to higher oxidative stability concerns, as evidenced by the higher TBARS content. Overall, each method exhibited distinct advantages and drawbacks, suggesting that the choice of extraction technique should be aligned with the specific quality parameters desired in the final oil product. Therefore, based on the higher yield, with lower cholesterol and trans-fatty acid content and higher lipid stability, wet rendering can be considered a simple, green, non-solvent extraction procedure and environmentally friendly method for safely extracting quality edible oil from pig brains, which may play an important role in expanding the use of by-products for the sustainable meat industry. However, the refinement process for improving the quality of pig brain oil extracted via the wet rendering technique could also be examined, and the application of pig brain oil in food and cosmetics products might be investigated further to determine its viability for industries.

## Figures and Tables

**Figure 1 foods-13-02818-f001:**
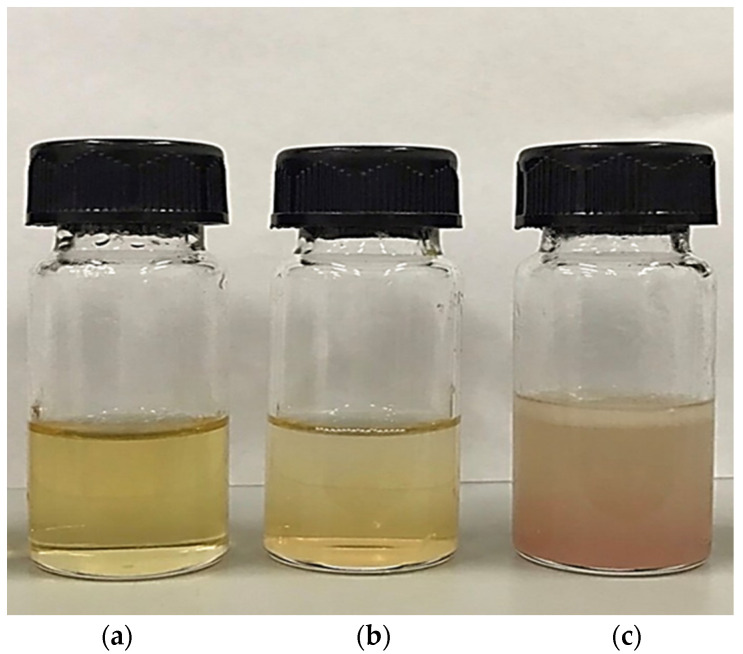
The appearance of pig brain oil obtained using various extraction methods, namely the Bligh and Dyer method (**a**), wet rendering (**b**), and aqueous saline (**c**).

**Figure 2 foods-13-02818-f002:**
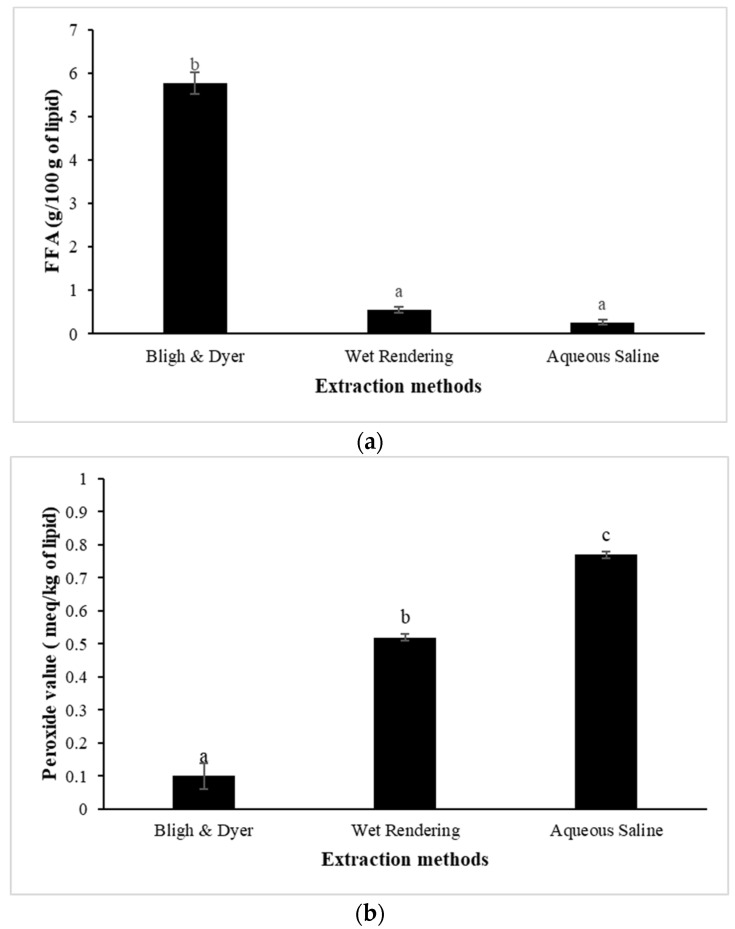
Free fatty acid (FFA); (**a**), peroxide value (PV); (**b**), and thiobarbituric acid reactive substances (TBARS); (**c**) of pig brain oil obtained using various extraction methods, namely Bligh and Dyer, wet rendering, and aqueous saline. The bars reflect the standard deviations of triplicate determinations. Different letters denote significant differences (*p* < 0.05).

**Figure 3 foods-13-02818-f003:**
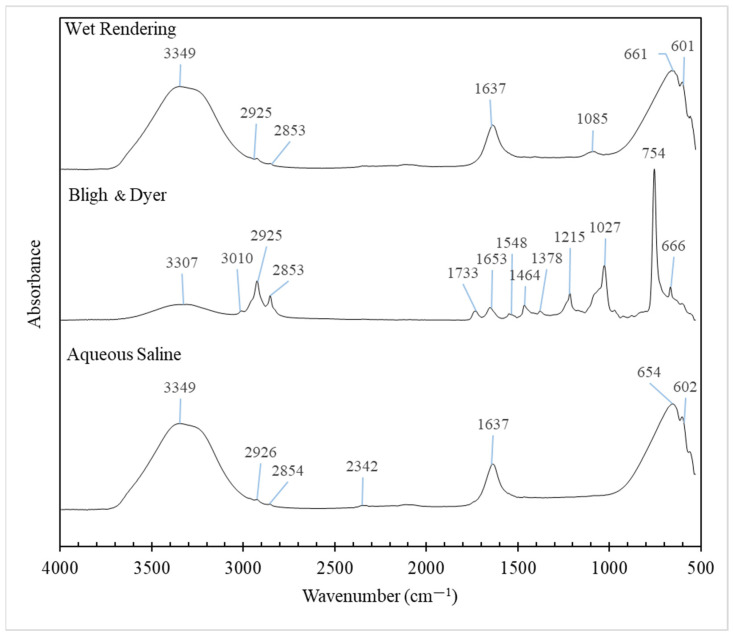
Fourier transform infrared (FTIR) spectra of pig brain oil obtained using various extraction methods, namely Bligh and Dyer, wet rendering, and aqueous saline.

**Table 1 foods-13-02818-t001:** Extraction yield and color of pig brain oil extracted by different methods.

Parameters	Extraction Method		
	Bligh and Dyer	Wet Rendering	Aqueous Saline
Extraction yield (%)	6.61 ± 1.03 ^b^	13.09 ± 0.18 ^c^	2.43 ± 0.30 ^a^
Color			
*L**	11.45 ± 0.69 ^c^	7.42 ± 0.16 ^a^	10.61 ± 0.14 ^b^
*a**	−0.20 ± 0.07 ^b^	−0.45 ± 0.28 ^a^	−0.06 ± 0.45 ^c^
*b**	0.28 ± 0.20 ^b^	0.27 ± 0.15 ^b^	−0.66 ± 1.39 ^a^
Redness index (*a**/*b**)	−0.71 ± 0.14 ^b^	−1.67 ± 0.21 ^a^	0.09 ± 0.12 ^c^

Values are shown as mean ± standard deviation of triplicate measurements. Significant differences (*p* < 0.05) are considered between letters in the same row.

**Table 2 foods-13-02818-t002:** Some lipid classes of pig brain oil extracted by different methods.

Compositions	Extraction Method		
	Bligh and Dyer	Wet Rendering	Aqueous Saline
Total phospholipid (g/100 g lipid)	3.22 ± 0.05 ^b^	0.08 ± 0.01 ^a^	0.11 ± 0.01 ^a^
Total cholesterol (mg /100 g lipid)	4305.70 ± 0.05 ^c^	45.65 ± 0.01 ^a^	173.72 ± 0.01 ^b^
Total carotenoids (mg/100 g lipid)	0.09 ± 0.90 ^c^	0.05 ± 0.88 ^b^	0.02 ± 0.27 ^a^
Total tocopherols (mg/100 g lipid)	66.24 ± 0.01 ^c^	33.07 ± 0.00 ^b^	27.35 ± 0.00 ^a^

Values are shown as mean ± standard deviation of triplicate measurements. Significant differences (*p* < 0.05) are considered between letters in the same row.

**Table 3 foods-13-02818-t003:** Fatty acid profile of pig brain oil extracted by different methods.

Fatty Acids (% of Total Fatty Acid)	Extraction Method		
	Bligh and Dyer	Wet Rendering	Aqueous Saline
Saturated fatty acid (SFA)			
Butyric acid (C4:0)	0.05 ± 0.02 ^c^	11.68 ± 0.40 ^a^	6.77 ± 1.44 ^b^
Caprylic acid (C8:0)	0.01 ± 0.01 ^b^	1.63 ± 0.24 ^a^	1.40 ± 0.38 ^a^
Capric acid (C10:0)	0.004 ± 0.00 ^b^	0.27 ± 0.16 ^a^	0.22 ± 0.03 ^a^
Undecylic acid (C11:0)	0.02 ± 0.00 ^b^	3.56 ± 0.78 ^a^	2.88 ± 0.41 ^a^
Lauric acid (C12:0)	0.005 ± 0.00 ^c^	0.38 ± 0.10 ^b^	0.51 ± 0.03 ^a^
Myristic acid (C14:0)	0.44 ± 0.30 ^b^	5.27 ± 0.43 ^a^	4.58 ± 0.79 ^a^
Pentadecanoic acid (C15:0)	0.08 ± 0.01 ^b^	0.87 ± 0.11 ^a^	0.74 ± 0.09 ^a^
Palmitic acid (C16:0)	27.26 ± 0.97 ^a^	21.50 ± 1.49 ^b^	19.17 ± 1.21 ^b^
Heptadecanoic acid (C17:0)	0.35 ± 0.02	nd *	nd
Stearic acid (C18:0)	16.03 ± 0.59 ^b^	39.79 ± 1.69 ^a^	35.95 ± 3.58 ^a^
Arachidic acid (C20:0)	0.19 ± 0.01 ^b^	1.02 ± 0.22 ^a^	nd
Heneicosanoic acid (C21:0)	0.01 ± 0.00	nd	nd
Behenic acid (C22:0)	0.08 ± 0.01	nd	nd
Tricosylic acid (C23:0)	0.47 ± 0.03	nd	nd
Lignoceric acid (C24:0)	0.06 ± 0.00	nd	nd
Total SFA	45.04 ± 1.31 ^c^	86.57 ± 2.26 ^a^	77.58 ± 3.70 ^b^
Monounsaturated fatty acid (MUFA)			
cis-10-pentadecenoic acid (C15:1)	12.93 ± 0.91	nd	nd
Palmitoleic acid (C16:1 *n*-7)	0.91 ± 0.04	nd	nd
Cis-10-heptadecenoic acid (C17:1)	0.23 ± 0.02	nd	nd
Elaidic acid (C18:1 *n*-9 trans)	20.64 ± 0.60 ^a^	11.51 ± 2.87 ^b^	18.22 ± 2.77 ^a^
Cis-11-eicosenoic acid (C20:1 *n*-11)	1.64 ± 0.07	nd	nd
Cis-13-docosenoate (Erucate) (C22:1)	0.33 ± 0.03	nd	nd
Cis-15-tetracosenoate (Nervonate) (C24:1)	1.74 ± 0.14	nd	nd
Total MUFA	38.41 ± 1.40 ^a^	11.51 ± 2.87 ^c^	18.22 ± 2.77 ^b^
Polyunsaturated fatty acid (PUFA)			
Cis-9,12-octadecadienoic acid (C18:2 *n*-6)	1.67 ± 0.06 ^b^	2.12 ± 1.92 ^b^	5.51 ± 4.20 ^a^
Cis-9,12,15-octadecatrienoic acid (C18:3 *n*-3)	0.03 ± 0.01	nd	nd
Cis-6,9,12-octadecatrienoic acid (C18:3 *n*-6)	0.04 ± 0.00	nd	nd
Cis-11, 14-eicosadienoic acid (C20:2 *n*-6)	0.57 ± 0.04	nd	nd
Cis-8, 11, 14-eicosatrienoic acid (C20:3 *n*-6)	0.67 ± 0.05	nd	nd
Cis-5, 8, 11, 14, 17-eicosapentaenoic acid (C20:5 *n*-3, EPA)	0.04 ± 0.01	nd	nd
Cis-13,16-docosadienoic acid (C22:2 *n*-6)	0.31 ± 0.39	nd	nd
Cis-4,7,10,13,16,19-docosahexaenoic acid (C22:6 *n*-3, DHA)	13.21 ± 0.97	nd	nd
Total PUFA	16.54 ± 0.66 ^a^	2.12 ± 1.92 ^c^	5.51 ± 4.20 ^b^

* nd: not detected. Values are given as mean ± standard deviation of triplicate measurements. Significant differences (*p* < 0.05) are considered between letters in the same row.

## Data Availability

The original contributions presented in the study are included in the article, further inquiries can be directed to the corresponding author.

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
