# Peer review of "Valorization of Pig Brains for Prime Quality Oil: A Comparative Evaluation of Organic-Solvent-Based and Solvent-Free Extractions"

_foods, 2024, doi:10.3390/foods13172818_

Round 1

Reviewer 1 Report

Comments and Suggestions for Authors

The study titled "Valorization of Pig Brains for Prime Quality Oil: A Comparative Evaluation of Organic Solvent-Based and Solvent-Free Extractions" examines various methods for extracting edible oil from pig brains, a by-product of the pork industry. It compares two non-solvent extraction techniques—wet rendering and aqueous saline—with the traditional solvent-based Bligh & Dyer method to identify the most effective approach for obtaining high-quality oil. The research evaluates factors such as oil yield, lipid classes, fatty acid profile, and lipid stability against lipolysis and oxidation.

Comments:

1. Statistical Analysis Results: All statistical analysis outcomes should be presented in superscript.

2. Fatty Acid Profile: Please add some discussion why the Bligh & Dyer method produces oil with a higher unsaturated fatty acid content compared to wet rendering and saline solution methods.

3. Peroxide value: add some information why peroxide value is higher in oils extracted through wet rendering and saline solution methods compared to the Bligh & Dyer method.

4. Extraction time. How long does the extraction process occur in all methods? Extraction time should be considered to have some impact on pig brain oil’s qualities.

Author Response

Reviewer#1

The study titled "Valorization of Pig Brains for Prime Quality Oil: A Comparative Evaluation of Organic Solvent-Based and Solvent-Free Extractions" examines various methods for extracting edible oil from pig brains, a by-product of the pork industry. It compares two non-solvent extraction techniques—wet rendering and aqueous saline—with the traditional solvent-based Bligh & Dyer method to identify the most effective approach for obtaining high-quality oil. The research evaluates factors such as oil yield, lipid classes, fatty acid profile, and lipid stability against lipolysis and oxidation.

Comments:

  1. Statistical Analysis Results: All statistical analysis outcomes should be presented in superscript.

Ans: Done.

  1. Fatty Acid Profile: Please add some discussion why the Bligh & Dyer method produces oil with a higher unsaturated fatty acid content compared to wet rendering and saline solution methods.

Ans: We originally stated that “The Bligh & Dyer method recovered the most fatty acids because it used solvent extraction to separate the majority of the lipid from brain tissue.”. Additionally, a new supportive remark was included. “The Bligh & Dyer extraction has long been recognized as an effective method for extracting triacylglycerols, FFA, and PL [16], making it possible to recover more unsaturated fatty acids from pig brain.

  1. Peroxide value: add some information why peroxide value is higher in oils extracted through wet rendering and saline solution methods compared to the Bligh & Dyer method.

Ans: A plausible explanation for the greater peroxide generation in aqueous saline and wet rendering compared to the Bligh & Dyer method was provided. “In the aqueous saline extraction, long-term stirring in water (30 min) followed by twice centrifuging for a total of 80 min may enhance peroxide production as compared to the Bligh & Dyer method, which used low temperature and short period homogenization (total of 3.5 min). During wet rendering, heating for 90 min may facilitate the formation of peroxide.”

  1. Extraction time. How long does the extraction process occur in all methods? Extraction time should be considered to have some impact on pig brain oil’s qualities.

Ans: The total processing time for each extraction method was specified in the Methods: approximately 45 min for the Bligh & Dyer method, 100 min for the wet rendering method, and 120 min for the aqueous saline approach.

Reviewer 2 Report

Comments and Suggestions for Authors

In this manuscript, the authors presented the results of a very interesting study. The aim of this research was to examine and compare different non-solvent techniques for extracting oil/fat from pig brains in comparison with the conventional solvent technique. The oil yield, fatty acid composition, physical characteristics, lipolytic and oxidative changes of the oil/fat were examined.

The research represents a significant contribution to the implementation of the “zero waste” concept in the meat industry, which are certain current trends and innovations of this research.

This is precisely what the authors should emphasize more in the objectives and conclusions of their research. Also, the possibilities for practical application of the obtained results should be mentioned and highlighted in this paper.

In addition, the following shortcomings should be addressed:

In the Materials and Methods section, part 2.4 should describe in more detail the determination of moisture content and oil/fat content. How were they used to determine the oil yield?

In section 2.5, it is not stated how the color was measured? Was the color measuring instrument previously calibrated and with what?

In section 2.7, when determining the fatty acid composition of the extracted oils, FAME standards were not used, only the MS database. Is this sufficient, given the very large differences in the fatty acid composition obtained by different extraction techniques?

In section 2.8, lipolytic and oxidative changes were not examined in sufficient detail. The section should be expanded with additional parameters, e.g., those indicating hydrolysis, lipolysis, formation of primary and/or secondary oxidation products.

In the Results and Discussion section, some results are not adequately discussed. This is the case with the results of determining the fatty acid composition of oils/fats obtained by extraction using different techniques. The drastic differences among these results should be explained and compared with any available data from the literature.

The conclusion is short, very general, and concise. It should be supplemented with the most significant results, innovations, and practical applications of the obtained results.

The very high degree of overlap (42%) of words in the text determined by the iThenticate software is concerning. The report shows that the overlap is mostly with the first 5 sources found. This should be addressed.

Author Response

Reviewer#2

In this manuscript, the authors presented the results of a very interesting study. The aim of this research was to examine and compare different non-solvent techniques for extracting oil/fat from pig brains in comparison with the conventional solvent technique. The oil yield, fatty acid composition, physical characteristics, lipolytic and oxidative changes of the oil/fat were examined.

The research represents a significant contribution to the implementation of the “zero waste” concept in the meat industry, which are certain current trends and innovations of this research.

This is precisely what the authors should emphasize more in the objectives and conclusions of their research. Also, the possibilities for practical application of the obtained results should be mentioned and highlighted in this paper.

Ans: Thank you very much for your insightful and significant suggestion. We did our best to revise the manuscript as recommended.

In addition, the following shortcomings should be addressed:

In the Materials and Methods section, part 2.4 should describe in more detail the determination of moisture content and oil/fat content. How were they used to determine the oil yield?

Ans: The moisture content was initially specified in Section 2.3.1 of the process to indicate that it was analyzed after each technique of extraction. “The moisture content of the oil sample was determined using a coulometric Karl Fischer titrator (C20, Mettler-Toledo Intl., Columbus, OH, USA).”

However, to explain the determination of the extraction yield, the statement in Section 2.4 was amended. “The yield was calculated using Equation (1), which employed the total weight of extracted oil after subtracting moisture content obtained using a coulometric Karl Fischer titrator.”

Extraction yield (%) = (Total weight of extracted oil - Moisture/Sample weight) × 100         (1)

In section 2.5, it is not stated how the color was measured? Was the color measuring instrument previously calibrated and with what?

Ans: The color measurement detail was improved as suggested. “Colorimetric values of pig brain oils, including L* (lightness), a* (red-ness/greenness), and b* (yellowness/blueness) were analyzed using a Hunterlab Color-Flex®EZ instrument (10° standard observers, illuminant D65; Hunter Assoc. Laboratory; Reston, VA, USA). Before performing color analysis, the equipment was calibrated using white and black standard plates. A glass sample cup was filled with oil samples of equal weight from each treatment and then analyzed for color. The redness index (a*/b*) was also reported.

In section 2.7, when determining the fatty acid composition of the extracted oils, FAME standards were not used, only the MS database. Is this sufficient, given the very large differences in the fatty acid composition obtained by different extraction techniques?

Ans: Actually, the standard curve was created, and the details were included in the revised manuscript. However, to avoid the similarity index in the Method section, the entire process can be obtained from the reference mentioned.

Fatty acid methyl esters (FAME) in the samples were measured with a gas chromatography/quadrupole time of flight (GC/Q-TOF) mass spectrometer (GC 7890B/MSD 7250, Agilent Technologies, USA) connected to the PAL auto sampler system (CTC Analytics AG, Switzerland). MS data was collected using Agilent Technologies' MassHunter software (version 10.0, Santa Clara, CA, USA). Myristic acid D27 (500 ppm in hexane) was used as an internal standard (IS). The calibration curves were created by combining an equivalent amount of FAME (20 to 1,000 ppm) with a solution of myristic acid-D27 methyl ester in hexane. The complete procedure as well as the optimal analytical condition can be retrieved from Chinarak et al. [23].”          

In section 2.8, lipolytic and oxidative changes were not examined in sufficient detail. The section should be expanded with additional parameters, e.g., those indicating hydrolysis, lipolysis, formation of primary and/or secondary oxidation products.

Ans:  Section 2.8. describes the procedures for lipolysis and lipid oxidation. As indicated below, the description to indicate what product was sought to be determined was initially given at the beginning of each analysis.

Lipolysis or lipid hydrolysis was monitored using the FFA content which was determined according to the method of Lowry and Tinsley [24].

The formation of primary lipid oxidation products was monitored using peroxide value (PV), according to the method described by Low and Ng [25].

The formation of secondary lipid oxidation products was determined using the thiobarbituric acid reactive substances (TBARS) assay according to the method described by Buege and Aust [26].

In the Results and Discussion section, some results are not adequately discussed. This is the case with the results of determining the fatty acid composition of oils/fats obtained by extraction using different techniques. The drastic differences among these results should be explained and compared with any available data from the literature.

Ans: The fatty acid analysis was carried out utilizing the accuracy and precision method and equipment, which included a gas chromatography/quadrupole time of flight (GC/Q-TOF) mass spectrometer and FAME standard curves. So, the fatty acid profile results were reliable.

The discussion was based on the obtained results, with the goal of explaining the data and relating it to the preceding study, which recorded several plausible hypothesized mechanisms. Because there had been no research on pig brain oil or other livestock brain oil, the discussion was based on available literature data. We attempted to connect with other accessible studies, such as tuna head or another source of oils prepared using similar or comparable processes.

The discussion was rechecked in the amended version, and additional details were provided in response to both reviewers' recommendations. Thank you very much.

The conclusion is short, very general, and concise. It should be supplemented with the most significant results, innovations, and practical applications of the obtained results.

Ans: A concise and informative conclusion was written to highlight the study's success, i.e. fulfilling the research objective. However, the most noteworthy findings, innovations, practical applications, and additional recommendations were presented.

The method of oil extraction from pig brains significantly influences the yield, composition, and stability of the oil. Wet rendering provided the highest oil yield, while the Bligh & Dyer method produced oil with the highest content of phospholipids, cholesterol, carotenoids, tocopherols, and unsaturated fatty acids, albeit with a higher trans-fatty acid content. The aqueous saline method resulted in the lowest yield and produced a discolored oil. Despite yielding the highest quality oil in some respects, the Bligh & Dyer method also led to higher oxidative stability concerns, as evidenced by the higher TBARS content. Overall, each method exhibited distinct advantages and drawbacks, suggesting that the choice of extraction technique should be aligned with the specific quality parameters desired in the final oil product. Therefore, based on the higher yield, lower cholesterol and trans-fatty acid content, and higher lipid stability, wet rendering can be considered a simple, green non-solvent extraction procedure, and environmentally friendly method for safely extracting quality edible oil from pig brains, which may play an important role in expanding the use of by-products for the sustainable meat industry. However, the refinement process for improving the quality of pig brain oil extracted via the wet rendering technique could also be examined, as well as the application of pig brain oil in food and cosmetics products might be investigated further to determine its viability for industry.”

The very high degree of overlap (42%) of words in the text determined by the iThenticate software is concerning. The report shows that the overlap is mostly with the first 5 sources found. This should be addressed.

Ans: The similarity index was decreased in accordance with both reviewer and editorial office guidelines. We attempted to decrease similarity and proofread the English using QuillBot, a paraphrase program.

Round 2

Reviewer 2 Report

Comments and Suggestions for Authors

Taking into account the comments of the reviewers, the authors have now significantly corrected the submitted manuscript.

The degree of overlap (similarity) of words in the text is reduced.